# A Finite Element Analysis of the Effects of Graphene and Carbon Nanotubes on Thermal Conductivity of Co Phase in WC–Co Carbide

**DOI:** 10.3390/ma14247656

**Published:** 2021-12-12

**Authors:** Zhengwu Li, Wenkai Xiao, Xuefeng Ruan

**Affiliations:** 1School of Power and Mechanical Engineering, Wuhan University, Wuhan 430072, China; lzw970630@163.com; 2Research Center of Nanoscience and Technology, Wuhan University, Wuhan 430072, China; xf-ruan@whu.edu.cn

**Keywords:** finite element law, cemented carbide, carbon nanotubes, graphene, thermal conductivity

## Abstract

In engineering practice, the service life of cemented carbide shield tunneling machines in uneven soft and hard strata will be seriously reduced due to thermal stress. When carbon nanotubes (CNTs) and graphene nano-platelets (GNPs) are added to WC–Co carbide as enhanced phases, the thermal conductivity of carbide is significantly improved. Research should be performed to further understand the mechanism of enhancement in composites and to find ways to assist the design and optimization of the structure. In this paper, a series of finite element models were established using scripts to find the factors that affect the thermal conduction, including positions, orientations, interface thermal conductivity, shapes, sizes, and so on. WC–Co carbide with CNTs (0.06%, 0.12%, and 0.18% vol.), GNPs (0.06%, 0.12%, and 0.18% vol.) and hybrid CNTs–GNPs (1:1) were prepared to verify the reliability of finite element simulation results. The results show that the larger the interface thermal conductivity, the higher the composite phase thermal conductivity. Each 1%vol of CNTs increased the thermal conductivity of the composite phase by 7.2%, and each 1% vol. of GNPs increased the thermal conductivity of the composite phase by 5.2%. The proper curvature (around 140°) of CNTs and GNPs with a proper diameter to thickness ratio is suggested to lead to better thermal conductivity.

## 1. Introduction

Carbide has the advantages of high hardness, high wear resistance, and good toughness for avoiding brittle fracture, and is a common material for shield blades. However, when a shield machine digs in the bottom layer of uneven soft and hard strata, the life of the tool will be seriously reduced due to thermal stress and other reasons. Beste, Coronel, and Jacobson et al. used focused ion beams and electron beams to work with thinning methods to prepare analytical samples of drill bit surface flakes and to perform TEM analysis of drill surface samples after the rock was drilled using a transmitted electron microscope [1,2]. Their research confirmed that the average operating temperature was 500 °C, but in extreme cases the instantaneous temperature of up to 1500 °C melted first as an adhesive at high temperatures, and the WC did not melt [2]. The current research shows that the main failure forms of carbide tools in shield machines are fracture and wear. The reason for failure is that, under the simultaneous action of mechanical stress and thermal stress, micro cracks are formed and then extended until the carbide tool fails [3,4]. The cause of carbide tool failure is due to thermal stress-induced cracking, but little research has been completed on how to improve the thermal conductivity of carbide to solve this problem. The addition of graphene and carbon nanotubes to cemented carbide may be a good solution to the low thermal conductivity of cemented carbide [5].

Carbon nanotubes (CNTs) have been widely used in various composite materials since their ability to improve their thermal properties was discovered. Multi-wall carbon nanotubes (MWCNTs) fillers reinforced to E-glass/Kevlar/epoxy composites (GKEC) are best suited for structural applications [6]. Carbon nanotubes are expected to be potential fillers for improving the thermal conductivity of epoxy composites due to their high aspect ratio and excellent thermal conductivity (4000 W/mK) [7,8].The same is true of graphene nano-platelets (GNPs). In addition to their high aspect ratio and superior thermal conductivity (5300 W/mK) [9], GNPs have a unique flat structure that provides a large heat-conducting surface, thus enhancing the thermal conductivity of composites better than CNTs [10,11]. Kuilla et al. have reported in detail on polymer composite graphene materials, which have improved electrical, mechanical and thermal conductivity to varying degrees [12]. By studying the thermal conductivity of graphene/polymer composites, the addition of graphene has been found to be able to significantly improve the thermal conductivity of polymers. However, there are problems associated with this, such as poor dispersion and the easy reunion of graphene materials in the substrate [13,14]. Graphene has also been added to carbide to improve the physical properties of carbide [15]. However, at present, there are still shortcomings in the study of the mechanism of adding phase to improve the thermal conductivity of carbide. If we can explore the influence of graphene and carbon nanotubes on carbide thermal conductivity from the perspective of simulation experiments, we can understand in essence how to improve the thermal conductivity of carbide.

With the increasing development of computer simulation technology, more and more research tends to quantitatively explore the effect of carbon nanotubes and graphene-related parameters on thermal conductivity, and tries to analyze thermal conductivity by law enforcement. The Engineering Faculty et al. modeled the thermal conductivity ratio of an antifreeze-based hybrid nanofluid containing graphene oxide (GO) and copper oxide (CuO) using mathematical methods to determine the effect of different volume fractions of GO on thermal conductivity [16]. Both molecular dynamics and finite elements are currently used for thermal simulation. Molecular dynamics rely mainly on Newtonian mechanics to model the motion of molecular systems in order to calculate the conformational integral of a system by taking samples in a system composed of different states of the molecular system. From a microcosmic point of view, the thermal conduction of GNPS/Co composite phase and CNTS/Co composite phase is mainly attributed to the vibrational motion of the acoustics, and molecular dynamics simulation can describe the vibration behavior of the acoustics in detail [17]. However, in continuum media mechanics, this interpretation referring to phonons cannot be used, so molecular dynamics is not applicable with this simulation case. Meanwhile, the finite element method (FEM) is considered a useful tool for estimating thermal conductivity, and models can also be established to show the heat flow of the added phase [18,19]. FEM can more fully reflect the internal conditions of composite materials through larger-scale simulation [20,21]. Long Nisheng et al. used the APDL method to simulate the changes in thermal behavior values in the laser metal deposition forming process, and the correctness of the simulation experiment was verified by experiments [22]. Huang Weibo et al. used FEM simulation technology to design millimeter-level simulation units to simulate the thermal behavior of laser cladding [23]. If this enhanced mechanism, and especially synergies, can be clearly explained, the microstructure design of composite materials can be better guided to achieve more reliable performance. However, to date, there have been no studies showing the factors that influence improvements in the thermal conductivity of carbide by graphene and carbon nanotubes.

Junjie Chen et al. introduced graphene oxide into metals and polymers and investigated the thermal properties of graphene oxide using nonequilibrium molecular dynamics, and the results showed that the degree of oxidation has a significant effect on the thermal conductivity. The oxygen-containing functional groups of graphene were found to have a detrimental effect on the thermal conductivity [24]. Hossein Ghaderi et al. evaluated the thermal conductivity of strained concentric multi-walled carbon and boron-nitride nanotubes and found that the length and diameter (morphology) of the nanotubes have an effect on their thermal conductivity [25]. Therefore, the morphology of the added phase (CNTs and GNPs), the volume fraction, and the thermal conductivity of the contact surface are all factors to be considered. This paper aims to establish a three-dimensional numerical model of the spatial random distribution of curved carbon nanotubes/Co phase composite phases, graphene/Co phase composite phase three-dimensional numerical models, and graphene and carbon nanotubes/Co phase composite phase three-dimensional numerical models. Further, it also aims to quantitatively explore carbon nanotube volume fractions, carbon nanotubes, and the Co phase effects of interface thermal conductivity and the carbon nanotube form on composite Co phase thermal conductivity, the effect of graphene volume fractions, graphene and Co-phase interface thermal conductivity, and graphene distribution on composite Co phase thermal conductivity, and the effect of graphene and carbon nanotubes on composite phase thermal conductivity. In this study, WO–Co carbide composites with hybrid fillings of CNTs, GNPs, and CNTs–GNPs were prepared. Thermal conductivity was tested, and the microstructures of the composite materials were studied.

## 2. Materials and Methods

### 2.1. The FEM Model Is Simplified

The thermal conductivity of cemented carbides with graphene and carbon nanotubes has been improved to some extent, and through EDS energy spectrum analysis it has been concluded that graphene and carbon nanotubes are mainly distributed in the Co phase of WO–Co carbide [26]. Youdi Kuang et al. used molecular dynamics to confirm that covalent functionalization has a significant effect on heat transport in carbon nanotube/polymer composites [27]. Additionally, the number of oxygen-containing functional groups of graphene affects the thermal conductivity of the graphene contact interface [24]. This indicates that the different nature of the added phases affects the contact thermal conductivity of the contact surfaces. However, WC–Co carbide is a continuous medium, and the contact thermal conductivity of the added phase with the Co phase cannot be simulated using molecular dynamics. Other methods are needed here to perform the simulation. Fan Tao et al. used a finite element method to simulate the thermal behavior of graphene nanoplatelets/epoxy composites, and then described the contact between graphene and epoxy resin with different properties by the concept of interfacial thermal conductivity [28]. This means that the thermal properties of these contact surfaces can be used as input parameters (contact surface thermal conductivity) for the FEM model in order to calculate their effective thermal properties, which greatly simplifies the calculation process. In this paper, a thermal computing model of GNPs/Co composite phase, CNTs/Co composite phase, and GNPs&CNTs/Co composite phase was established by using ANSYS Workbench, a large piece of commercial FEM software. A composite model with different GNP or CNT volume fractions was established, and several parameters were controlled to study the thermal behavior of composite materials. In the FEM, the discrete and complex geometry of randomly distributed GNPs/epoxy composites can result in high computational costs. Based on the following assumptions and simplifications, this problem can be mitigated with minimal loss of accuracy:WC–Co carbide is made by bonding the hard phase WC and the bonding phase Co. The addition of GNPs and CNTs enhances the thermal conductivity of cemented carbide. However, GNPs and CNTs are not uniformly distributed throughout the cemented carbide, but are rather concentrated in the Co phase or on the contact surface between the Co phase and WC. Therefore, the FEM model for making WC–Co carbide can be simplified to make a finite element model of the Co phase with the addition of GNPs and CNTs;All GNPs are reduced to cylinders with uniform diameters at the nanoscale, and all CNTs are reduced to even length graphene cylinders at the nanoscale, creating a two-dimensional nanomaterial. The principle of minimum surface energy shows the natural states of three-dimensional surfaces. In addition, graphene is much more rigid than one-dimensional carbon nanotubes. Therefore, GNPs can be thought of as discs with a certain diameter and thickness to simplify the modeling process;It is assumed that the nature of GNPs and CNTS is temperature independent. In fact, the performance of GNPs and CNTS depends on temperature. If they are known to be dependent on temperature, they can be taken into account. However, in order to simplify the modeling process, temperature dependence is not overlooked in this work.

### 2.2. FEM Model

Taking into account our existing computing resources, we defined a representative volume unit called represent volume active element (RVE) (shown in Figure 1a) with a cube cell size of 50 × 50 × 50 nm. The nature of GNPs, CNTS, and Co phases is shown in Table 1. Using ANSYS Workbench software (Version 18.0), a computational model was built using scripts in SpaceClaim. Scripts allow models to be built quickly. The parameter adjustment is accurate and the calculation is repetitive. The model was set up as follows:

#### 2.2.1. The Establishment of the GNPS Model

This model has regular geometric parts. When modeling a cylinder using the script in SpaceClaim, three feature points are needed to generate the cylinder at the coordinate origin, as shown in Figure 1c. The diameter and thickness of GNPs are determined according to the specific experimental content. Three sets of 0–50 random numbers are generated, corresponding to the displacement of graphene from the origin X, Y, and Z directions. Similarly, three sets of 0–180 random numbers are generated, corresponding to the rotation angle of each unit of graphene with x-, y-, and z-axes as the axis of rotation. After calculation, the spatial location of the generated GNPs can be determined and the data summarized into a characteristic point data group for follow-up studies.

#### 2.2.2. The Establishment of the CNTs Model

This model has regular geometric parts. Using the SpaceClaim model script, a curved CNT cylinder is created by sweeping through a round surface along the feature curve, which consists of three feature points, as shown in Figure 1b. However, the curved cylinder obtained by sweeping can only be generated from feature curves, so the displacement and rotation of CNTs are added to determine the specific spatial position of CNTs. The diameter and length of CNTs are set, according to the specific experimental content.

#### 2.2.3. The Establishment of the RVE Model of Composite Materials

To realize the automatic generation of the composite RVE geometric model, the morphology of GNPs and CNTs, the distribution of GNPs and CNTs in the substrate, and the degree of interference between GNPs and CNTs and the substrate must be determined. The automatic generation algorithm flowchart for the composite RVE model is shown in Figure 2.

The key to achieving a spatially randomly distributed FEM model is to ensure that the spatial position and rotation angle of each GNP or CNT are randomly distributed. However, in order to make the model more realistic, adding phases should satisfy the case of no spatial overlap. In the following, the RVE model with GNP addition is used as an example to describe the process of building the FEM model. In the first step, three sets of random numbers ranging from 5 to 45 were generated, with an unlimited number of each set used. In the order of generation, one number was taken from each group and set as (x_i_, y_i_, z_i_) as the 3D spatial coordinates of the feature point of a single GNP. The reason for setting the range of random numbers from 5 to 45 is that the size of the RVE model is 50, and placing GNPs at the edge of the model will result in an excised volume that is too large, thus increasing the number of GNPs in the RVE model significantly. Too many small volume models can greatly increase the calculation time of FEM simulation and reduce the accuracy of the simulation. Then, three sets of random numbers were generated from −180 to 180, with an unlimited number of each set used. In the generated order, one number from each group was taken and set as (α_i_, β_i_, γ_i_) as the rotation angle of individual GNPs. α_i_ is the angle of rotation of the model around the x-axis, β_i_ is the angle of rotation of the model around the *y*-axis, and γ_i_ is the angle of rotation of the model around the *z*-axis. In the second step, the feature points that did not meet the constraints were removed by setting the constraints between the GNP feature points. The resulting array of feature points was then imported into SpaceClaim’s script to automatically generate all individual graphene models. Finally, GNPs beyond the RVE model boundary were cropped and the entire RVE model was properly adjusted to meet the desired volume fraction ratio. In particular, the relevant parameters of the rotated CNTs were calculated simultaneously during the process of CNT model building to facilitate the subsequent experiments.

#### 2.2.4. Consider the FEM Method of Thermal Resistance

The thermal resistance in this experiment was an important factor that directly affected the thermal conduction performance of the RVE thermal model, and the thermal conduction module in the ANSYS Workbench platform of the commercial FEM software was used for thermal conduction analysis. In the experiment, the thermal resistance of GNPs and Co phase substrates was defined as interface thermal conduction C_G-Co_, and the thermal resistance of CNTs and Co phase substrates was defined as interface thermal conduction C_C-Co_. Finally, the thermal resistance between GNPs and CNTs was defined as contact thermal conduction C_G-C_. The contact thermal conduction of carbon nanotubes is related to the interface of the contact material. Huxtable et al. immersed one-arm carbon nanotubes into the octane melt analog contact interface and demonstrated that the interface thermal conduction is related to the length, diameter, etc., of CNTs [29]. Using the idea of the controlling variable method, the parameterized thermal resistance dataset the values in the thermal conduction simulation experiment, and these values were related to RVE’s composite thermal conductivity for analysis, forming a one-to-many correspondence. Graphical representations of these parameters’ influence on the thermal conductivity of the composites could therefore be more intuitively represented.

#### 2.2.5. Thermal Conductivity Calculation

In order to accurately capture changes in the thermal flow network, in this study, heat flow and thermal conductivity were concentrated on a single axis. As shown in Figure 3, a constant temperature and a constant heat flow were set on the opposite sides of the square cell. The other four sides of the cube were adiabatic to ensure that heat flowed in one direction.

The thermal conductivity (k) of the composite material was calculated according to the Fourier equation:(1)$$k=\frac{q_{z}\Delta_{z}}{T_{Z+}-T_{Z-}}$$
where q_z_ is the constant heat applied on the face, Δz is the size of the matrix, T_z+_ is the average temperature of the face where heat flux was imposed, and T_z−_ is the average temperature of the face where the constant temperature was imposed.

Instead of changing the set parameters, such as volume fraction and morphology, only changing the random array of carbon nanotubes and graphene leads to the obtaining of data which meet the experimental conditions. Average values were taken to eliminate FEM simulation errors.

### 2.3. Validation Experiments

#### 2.3.1. Materials and Pretreatment

In this study, tungsten carbide powder (XFNANO, Nanjing, China) with a particle size of 1 μm and a purity of more than 99.9% was used. Nanocobalt powder with a particle size of 100 nm, a purity of 99.9%, a surface area of 40 to 60 m^3^/g, and a density of 8.9 g/cm^3^ was used. The single-walled carbon nanotubes used were 0.8 nm thick and 0.5–2 μm in diameter, and had about a 99% purity, an 80% single-layer rate and 5800 W/(m·K) heat conductivity. The specific surface area of the reagent grade single-layer graphene was 380 m^2^/g, and it had a purity greater than 90% and a thermal conductivity of 4000 W/(m·K). The controlled variable method was chosen to conduct the experiment. Seven control groups were set up in the experiment, adding CNT (0.06%, 0.12%, 0.18% vol.), GNP (0.06%, 0.12%, 0.18% vol.), and CNT (0.06% vol.) & GNP (0.06% vol.) cemented carbide samples, respectively. The composition design of the samples is shown in Table 2, with three samples included in each group. Each sample in each group was measured three times; in other words, each recorded value was the average of three measurements. G1-C1 was compared with the measured data of G2 and C2 to verify whether the GNPs and CNTs had a synergistic effect in improving the thermal conductivity of the materials.

The ultrasound of the graphene and carbon nanotubes was conducted before the mixed powder was prepared. Graphene and carbon nanotubes were added to acetone, mechanical stirring took place for 10 min, and then ultrasonic processing took place for 60 min to complete the dispersion of graphene and carbon nanotubes. The ultrasonic instrument came from China (Nanjing) Kunshan Ultrasonic Instrument Co., Ltd. production (product number KQ2200B). The mixing solution after the ultrasound was stirred at 100 °C until acetone was completely volatile. In the experiment, according to the distribution ratio in Table 2, the powder mixing and dispersion were carried out using the line planet mill; the fixed ball grinding speed was 110 rpm, the ball grinding time was 3.5 h, and ethanol was used as the medium [31,32]. The mixed powder was dried in a vacuum environment using a vacuum drying furnace. The dried powder was poured into a high-strength graphite mold, and a cylindrical specimen with a diameter of 20 mm was prepared with SPS sintering equipment (Wuhan University of Technology, Wuhan, China). The sintering temperature was 1200 °C, the sintering pressure was 80 MPa, and the insulation time was 10 min [33,34]. Because the sample was too hard, diamond stone grinding sheets were used to smooth the upper and lower surfaces of the sample. Using wire cutting technology, the sintered sample was cut into a 10 × 10 mm^2^ square block and two 10 × 2 mm^2^ striped samples. Among them, square samples were used as thermal diffusion coefficients and for hardness measurement and observation morphology, and long strip samples were used for the bend strength test and observation of the broken shape.

#### 2.3.2. Experimental Contents

This experiment did not meet the conditions for direct measurement of the thermal conductivity of the material due to the sample size. The thermal conductivity value was calculated by measuring the thermal diffusion coefficient and other parameters. Using the TC-7000H laser thermal constant tester (Wuhan University of Technology, Wuhan, China), the thermal diffusion coefficient of WC–6Co carbide samples at 400 °C was determined by a laser method. The thermal capacity of WC–Co carbide samples at 400 °C was measured using differential scanning volume heat (DSC) (Wuhan University of Technology, Wuhan, China). The sample quality was measured using electronic scales, and the Archimedes method was used to measure the sample volume and to calculate the sample density. Each sample was measured more than three times, and the final average was recorded in the data table. A formula was used to calculate the thermal conductivity of the samples.

## 3. Results and Discussion

### 3.1. FEM Simulation

In the meshing of the finite element simulation, the RVE model substrate used a tetrahedral mesh of type C3D10 with an element size of 1.2 × 10^−3^. Both the GNP and CNT models used a hexahedral mesh of C3D20 with an element size of 8 × 10^−4^. The mesh refinement of the contact surface was also set. All contact surfaces of the GNPs and CNTs with the Co phase were set up in thermal contact.

#### 3.1.1. Effect of CNTs on Thermal Conductivity of Carbide Composites

In order to explore the influence of carbon nanotubes on the thermal conductivity of carbide composites, a representative volume unit of 50 × 50 × 50 nm^3^ was set; the substrate of the unit was the Co phase and the carbon nanotube was the added phase. In the experiment, the parameters of the carbon nanotubes were set by default as the following: a radius of 0.8 nm, a length of 40 nm, and a bending degree of 140 degrees. Changes to the carbon nanotube morphology as specific conditions were explored. For example, Figure 4 presents a FEM simulation model with a volume fraction of 2% of a carbon nanotube.

1.The effects of C_C-Co_ on K_c_

Carbon nanotubes have different amounts of thermal conductivity because of the number of layers of the tube wall, morphology, factors of point defects, etc. The interfacial thermal conductivity C_C-Co_ was used to define the heat transfer efficiency between the carbon nanotubes and the Co phase matrix. A random model (0.5%, 1%, 2%, 3% vol.) was established based on the FEM. The performance of the CNTs and Co substrates used in the model is shown in Table 1. As shown in Figure 5, the thermal conductivity K_c_ of the composite materials was the result of calculating the different volume fractions of different thermal conductivity C_C-Co_ (1 × 10^3^, 1 × 10^4^, 1 × 10^5^, 1 × 10^6^, 1 × 10^7^, 1 × 10^8^, 1 × 10^9^, 1 × 10^10^ W/(m^2^·K)) and carbon nanotubes with different interfaces. Each 1% vol. of CNTs increased the thermal conductivity of the composite phase by 7.2%.

The carbon nanotubes had a very high thermal conductivity, and each carbon nanotube formed a thermal influence zone around the Co phase substrates, each of which, in turn, interacted with each other. Other carbon nanotubes located around the carbon nanotube thermal influence zone were able to quickly transfer heat through this thermal influence zone. If all the clustered heat affected areas were connected to each other, a thermal conduction network was able to be formed. The efficiency of the thermal conduction network determined the thermal conductivity of the composite materials, and the efficiency of the thermal conduction network was evaluated by the distribution and quantity of the carbon nanotubes. Under the condition of the interface thermal conductivity being unchanged, the thermal conductivity gradually increased with increases in the volume fraction. The thermal conductivity increment of the volume fraction was approximated, which could be explained with the conjecture of the heat conduction network, which was also proposed by Kwon et al., at CNT loadings of 0.2–1.4% vol. [35]. There was a critical value for the interfacial thermal conductivity. When the interfacial thermal conductivity was lower than the critical value, increases in the volume fraction were found to decrease the thermal conductivity of the composites instead.

Commonly used carbon nano-control preparation methods are: the arc discharge method, chemical vapor deposition method, laser evaporation method, template method, and so on. According to the number of layers that make up the walls of carbon nanotubes, carbon nanotubes can be divided into single-walled carbon nanotubes and multi-wall carbon nanotubes [36]. In addition to the types of carbon nanotubes, the interface thermal conduction of carbon nanotubes is also related to the purity, defects, and contact of carbon nanotubes [37,38]. At an interface thermal conductivity of 1 × 10^3^ to 1 × 10^6^ W/(m^2^·K), the thermal conductivity was found to increase as the interface thermal conductivity increased. However, when the interface thermal conductivity was greater than 1 × 10^8^ W/(m^2^·K), the thermal conductivity tended to stabilize with the improvements in the interface thermal conductivity.

2.The effects of the morphology of carbon nanotubes on K_c_

In addition to the layers of carbon nanotubes, the morphology of carbon nanotubes is also determined by the bending degrees, radii and lengths of carbon nanotubes. Without changing the lengths and radii of carbon nanotubes, five kinds of carbon nanotubes with different bending degrees were set, and the bending angles were 30 degrees, 60 degrees, 100 degrees, 140 degrees, and 180 degrees, respectively. A single carbon nanotube model is shown in Figure 6.

As can be seen from Figure 7a, the bending degrees of the carbon nanotubes had a certain effect on thermal conductivity. The thermal conductivity of the composite materials was better at a bending angle of 140 degrees, and the thermal conductivity was poor at a bending angle of 30 degrees.

After exploring the effect of carbon nanotube bending on the thermal conductivity of composite materials, the curve angle of the carbon nanotubes was set to 140 degrees, with the length being 40 nm, and four sets of carbon nanotube models with different radii were set. The carbon nanotubes had radii of 0.5 nm, 1 nm, 1.5 nm, and 2 nm, respectively.

As can be seen from Figure 7b, under the same volume fractions, the smaller the radii of the carbon nanotubes, the larger the contact areas between the carbon nanotubes and Co phase, and the greater the influence factors of the interfacial thermal conductivity. Under the same amount of interfacial thermal conductivity, the larger contact areas were also more conducive to heat flow, and the thermal conductivity of the composites was higher.

3.The effects of the dispersion of carbon nanotubes on Kc

According to the previous discussion, the heat-affected zone plays a very important role in heat conduction. The dispersion and connection of the heat-affected zone have a great influence on the thermal conductivity. In order to prove this conjecture, a characteristic parameter was tried to roughly describe the influence of the dispersion mass thermal conductivity in the heat-affected zone. Luo et al., proposed a dispersion coefficient D to effectively quantify the dispersion of inclusions in mixed microstructures [39]. D is defined according to the frequency distribution of the space point spacing data fitted by polynomial f(x), which is calculated with Formula (2), and μ is defined as the probability that the space point free path spacing falls into a certain range of the average spacing. μ is calculated with Formula (3).
(2)$$f\left(x\right)=\sum_{i=0}^{n}a_{i}x^{i}$$
(3)$$\mu =\frac{\sum_{i=0}^{n}\sum_{j=i+1}^{N}R_{\mathrm{ij}}}{N\left(N-1\right)/2}$$
where R_ij_ is the i_th_ carbon nanotube body center space coordinate and the j_th_ carbon nanotube body center space coordinate in the model, and N is the number of carbon nanotubes.

D_0.1_ and D_0.2_ are defined as all the distances between carbon nanotubes μ ± 0.1 μ and μ ± 0.2 μ possible within the scope. The D_0.1_ and D_0.2_ are calculated with Formulas (4) and (5).
(4)$$D_{0.1}=\int_{0.9\mu }^{1.1\mu }f\left(x\right)\mathrm{dx}$$
(5)$$D_{0.2}=\int_{0.8\mu }^{1.2\mu }f\left(x\right)\mathrm{dx}$$

The greater the D_0.1_ and D_0.2_ values [39], the more concentrated the distance between carbon nanotubes in space. The more carbon nanotubes are evenly distributed, the better the dispersion quality of carbon nanotubes in the whole RVE model.

Four models with a CNT volume content of 2% and different dispersion states were established. At the same time, the three conditions of carbon nanotube length, radius, and interfacial thermal conductivity were set as unchanged to eliminate the influence of these three parameters on the thermal conductivity of the model. When each model generated CNTs, the coordinates of three characteristic points of each carbon nanotube were recorded, and on this basis, the spatial coordinates, D_0.1_ and D_0.2_ of the body centers of all CNTs after rotation were programmed and calculated in MATLAB software. Each model had independent generation parameters and used the same generator code written in SpaceClaim.

The calculation results are shown in Table 3. The more carbon nanotubes were dispersed, the greater the values of D_0.1_ and D_0.2_, and the higher the thermal conductivity of the composite. The thermal conductivity of carbon nanotubes and metal co was very different. If heat is compared to water, metal co is equivalent to soil, and carbon nanotubes are equivalent to water pipes. The more evenly distributed the water pipes in the soil, the faster the water flows through the soil.

#### 3.1.2. Effect of GNPs on Thermal Conductivity of Carbide Composite

1.The effects of C_G-Co_ on K_c_

Graphene is generally prepared by acid oxidation, mechanical stripping or chemical vapor deposition. Therefore, the interfacial thermal conductivity of graphene prepared by different methods is different from that of Co phase. The number of oxygen-containing functional groups on graphene affects its thermal conductivity [24]. It is necessary to discuss the influence of interfacial thermal conductivity C_G-Co_ on Kc and establish models with different interfacial thermal conductivity C_G-Co_. In these models, graphene was randomly dispersed in the model with the same shape (radius 5 nm, thickness 1 nm). Figure 8 shows an example of a 2% volume fraction FEM model of graphene. 

In the randomly distributed GNP/Co material model, the influence of interfacial thermal conductivity C_G-Co_ on K_c_ is shown in Figure 9. When the interfacial thermal conductivity C_G-Co_ was greater than 4 × 10^4^ W/(m^2^·K), the thermal conductivity of K_c_ was always greater than that of the matrix Co (100 W/(m·K). Each 1% vol. of GNPs increased the thermal conductivity of the composite phase by 5.2%. As the interfacial thermal conductivity increased, the thermal conductivity of the composites increased further. Critical interfacial thermal conductivity existed when the interfacial thermal conductivity C_G-Co_ was less than 4 × 10^4^ W/(m^2^·K); however, the addition of graphene decreased the thermal conductivity K_c_ of the material. When the interfacial thermal conductivity C_G-Co_ was greater than 4 × 10^7^ W/(m^2^·K), K_c_ tended to stabilize. It can also be seen from the diagram that K_c_ was almost linearly related to C_G-Co_ when the volume fraction of graphene was unchanged.

2.The effects of the diameter–thickness ratio of graphene on K_c_

Different models were established by only changing the thickness of graphene. The volume fraction of graphene was set at 1%, and the interfacial thermal conductivity C_G-Co_ was set at 1 × 10^8^ W/(m^2^·K) with a radius of 5 nm. By changing the thickness of graphene, the diameter–thickness ratio of graphene can be changed, and the effect of the diameter–thickness ratio of graphene on the thermal conductivity of composites can be explored. As shown in Figure 10, the thinner the graphene sheet, the larger the contact area and the higher the thermal conductivity of the composites with the same volume fraction of graphene.

3.The effects of Agglomeration on K_c_

GNPs tend to agglomerate due to interlaminar Van der Waals forces. The agglomeration of GNPs has a very negative effect on the thermal conductivity of epoxy resin composites. One important reason why GNPs do not give full play to their excellent thermal properties is that they cannot solve the agglomeration problem fundamentally. It is impossible to establish a stochastic FEM model according to the aggregation state of graphene in reality, so the aggregation behavior of graphene was simplified to the superposition state of several graphene sheets, and the parameter of contact thermal conductivity between the graphene sheets was increased to restore the aggregation of graphene in real life as far as possible. In the model, the volume fraction of graphene was set to 1%, the ratio of diameter to thickness was 5, the interfacial thermal conductivity was 1 × 10^8^ W/(m^2^·K), and the release thermal conductivity between the graphene sheets was 1 × 10^11^ W/(m^2^·K). The set contact thermal conductivity of graphene is calculated by molecular dynamics. Because agglomeration in the simulation model is determined by the number of overlapping graphene sheets, it was found that as the number of overlapping sheets increased, the bad contact area between the graphene sheets increased greatly, and the favorable interface area between graphene and matrix Co phase decreased greatly, resulting in a significant decrease in thermal conductivity. The results are shown in Figure 11.

#### 3.1.3. Effect of CNTs and GNPs on thermal conductivity of cemented carbide composites

On the basis of previous research, we wanted to explore whether co-conduction occurs when graphene and carbon nanotubes are added together. A model with 1% volume fraction graphene and 1% carbon nanotubes was set up, while the control group had 2% volume fraction graphene and 2% volume fraction carbon nanotube models. Figure 12 presents one of several calculations for this model, with the setting parameters as follows: interface thermal conductivity C_C-Co_ and C_G-Co_ were both 1 × 10^8^ W/(m^2^·K), there was a 5 nm diameter–thickness ratio of graphene, a 40 nm length, 1 nm radius, and 140 degree bending angle of the carbon nanotubes. Graphene and carbon nanotubes were randomly distributed in the Co phase.

4.The effects of C_G-C_ on K_c_

It can be seen from Figure 13a that the interface thermal conductivity of GNPs–Co and CNTs–Co had a greater influence on the thermal conductivity than that of GNPs–CNTs. Changing the contact thermal conductivity only changed the thermal conductivity from 113 to 114 W/(m·K). Although the contact thermal conductivity of graphene with carbon nanotubes was greater than that of interface thermal conductivity C_C-Co_ and interface thermal conductivity C_G-Co_, the thermal conductivity between graphene and carbon nanotubes must be better than that between carbon and metal. However, the contact area between the graphene and carbon nanotubes was very different, being only 50 to 100 nm^2^, while the contact area between the reinforcement phase and matrix Co was 3000 nm^2^.

Additionally, it can be seen from Figure 13b that the thermal conductivity of the composites did not change much even when the contact area between the graphene and carbon nanotubes was changed.

5.Effective heat flow length

When exploring the influence of the dispersion of carbon nanotubes on thermal conductivity independently, the dispersion coefficient D was used to describe the dispersion of the enhanced phase in the system. However, the spatial difference between graphene and carbon nanotubes could not be reflected, because the calculation of dispersion coefficient D is based on the body-centered coordinate. When graphene is represented in body-centered coordinates, the heat-affected zone is limited in its longitudinal length. When carbon nanotubes are represented by the same body-centered coordinate, the heat-affected zone has a long length but a small area on the plane. Considering these factors comprehensively, the effective heat flow length L was determined to describe the influence of carbon nanotubes and graphene on thermal conductivity in the model.
(6)$$L_{G}=\sum_{i=0}^{N}L_{i}$$
(7)$$L_{C}=\sum_{j=0}^{M}L_{j}$$
(8)$$L=L_{G}+L_{C}$$

L_i_ represents the projection length of the i_th_ graphene in the direction of heat flow, L_j_ represents the projection length of the j-root carbon nanotube in the direction of heat flow, L_G_ represents the effective heat flow length of graphene, L_C_ represents the effective heat flow length of the carbon nanotube, and L represents the total effective heat flow length in the RVE model.

According to Formulas (5)–(7), the values of each model were calculated and the average number was taken to eliminate the error, with the results being shown in Table 4. The longer the effective heat flow length, the longer the heat flow could pass through the reinforcing phase, just as the longer the water tube length in the soil, the faster the water flow.

6.Cooperative thermal conductivity

Graphene and carbon nanotubes have different shapes in the heat-affected zone of Co. Using the characteristics of their heat-affected zones, we assumed that if graphene and carbon nanotubes were evenly dispersed in the model, a tighter heat-affected zone could be formed, which would further improve the thermal conductivity of the material. The thermal conductivity of the RVE model with 1% GNPs and 1% volume fraction CNTs was calculated according to the above model, and was compared with the RVE model with 2% volume fraction graphene and the RVE model with 2% volume fraction CNTs. The results are shown in Figure 14.

The results show that no conjectured synergistic thermal conductivity occurred and that the carbon nanotubes had a better thermal conductivity improvement than graphene. This may be because in real life, the same carbon filler tends to agglomerate due to Van der Waals forces, but in this study, both models were idealized because the carbon filler was evenly dispersed in the matrix. At the same time, because the carbon nanotubes in the model were slender, they had a larger contact area with the matrix at the same volume fraction.

### 3.2. Validation Experiments

It should be noted that in this experiment, the sizes of the cemented carbide samples sintered by spark plasma sintering did not match the standards for the direct measurement of the thermal conductivity of materials. Therefore, the thermal conductivity of cemented carbide could only be calculated laterally by measuring other physical parameters. By referring to the research of Chen et al. [40] on the theoretical calculation of the equivalent thermal conductivity of composite materials, thermal conductivity can be calculated by the following formula:(9)$$K_{c}={\alpha C}_{p}\rho$$
where K_c_ is the thermal conductivity, α is the thermal diffusivity, C_p_ is the heat capacity, and ρ is the density. Consequently, it was necessary to measure the thermal diffusivity, density and specific heat capacity of the cemented carbide samples.

The thermal diffusion test results for the WC–Co carbide specimens are shown in Table 5. After testing the thermal diffusion coefficient, the following data were obtained at 400 °C. As shown in Table 6, the volume and mass of each sample and the average density of the sample group were calculated. The sample volume was measured by the Archimedes drainage method, the sample mass was measured by electronic scale, and the sample density was calculated. The specific heat capacity of the cemented carbides at 400 °C is shown in Table 7.

The calculation results for the thermal conductivity of the cemented carbide are shown in Table 8. The thermal conductivity of the cemented carbide samples with 0.06% vol. carbon nanotubes and 0.06% vol. graphene increased by 11.7% compared with the original samples. The thermal conductivity values of the cemented carbide samples with 0.06% vol., 0.12% vol., and 0.18% vol. carbon nanotubes were 0.03%, 9.5%, and 12.2% higher than those of the original samples, respectively. The thermal conductivity values of the cemented carbide samples with 0.06% vol., 0.12% vol., and 0.18% vol. graphene were 11.3%, 14.3%, and 15.7% higher those that of the original samples. It is noteworthy that the thermal conductivity of the samples with 0.18% vol. graphene significantly increased by about 16%, more significantly than that of the CNTs with the same content. The experimental results correlate well with those of the finite element simulations, which further validates the feasibility of the finite element simulation experiments. Additionally, because of the non-single nature of the experimental factors, the control variables method could not be used, so the experimental results could only match the simulation results in terms of trend. It should be noted again that the thermal conductivity of the cemented carbides in this experiment was calculated. Therefore, the errors of the three measured parameters led to deviations between the calculated values and the actual values. In other words, it would be passable if the trend of the thermal conductivity of the different cemented carbide samples was similar to that of the FEM simulation.

## 4. Conclusions

In this paper, a programmable controlled FEM model with a random distribution of the added phases in space was innovatively developed, and the factors affecting the mixing of carbon nanotubes and graphene on the thermal conductivity of WC–Co cemented carbide were discussed. Sintering experiments were designed to verify the correctness of the results of the FEM in terms of trend. It can be seen from the results of FEM simulation that:The better the dispersion of CNTs, the larger the volume fraction, the smaller the diameter and the greater the interfacial thermal conductivity with CO phase, the higher the thermal conductivity of composites. Each 1% vol. of CNTs increased the thermal conductivity of the composite phase by 7.2%. An appropriate degree (about 140°) of bending will provide better results;The better the dispersion of GNPs, the larger the volume fraction and the ratio of diameter to thickness, and the greater the interfacial thermal conductivity with CO phase, the higher the thermal conductivity of composites. Each 1% vol. of GNPs increased the thermal conductivity of the composite phase by 5.2%. The dispersed graphene creates a wider range of heat flow networks in space;When CNTs and GNPs were incorporated into the Co phase, the magnitude of the contact thermal conductivity C_C-G_ had less effect on the thermal conductivity of the composite phase due to the small contact area between CNTs and GNPs.

All of the above factors can be improved to increase the thermal conductivity of carbide with the addition of CNTs and GNPs, which was verified in this study with sintering experiments.

## Figures and Tables

**Figure 1 materials-14-07656-f001:**
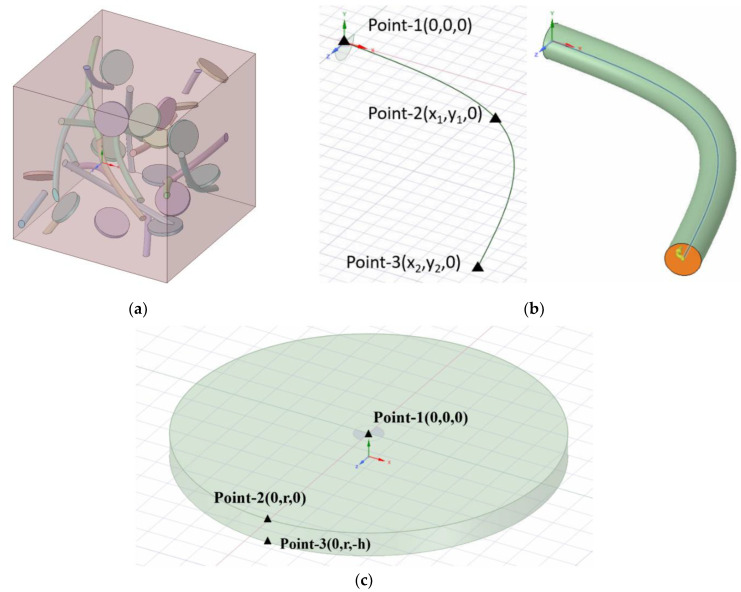
Schematic diagram of RVE model building. They should be listed as: (**a**) RVE finite element model; (**b**) A model of a filler (Carbon nanotubes); (**c**) A model of a filler (Graphene).

**Figure 2 materials-14-07656-f002:**
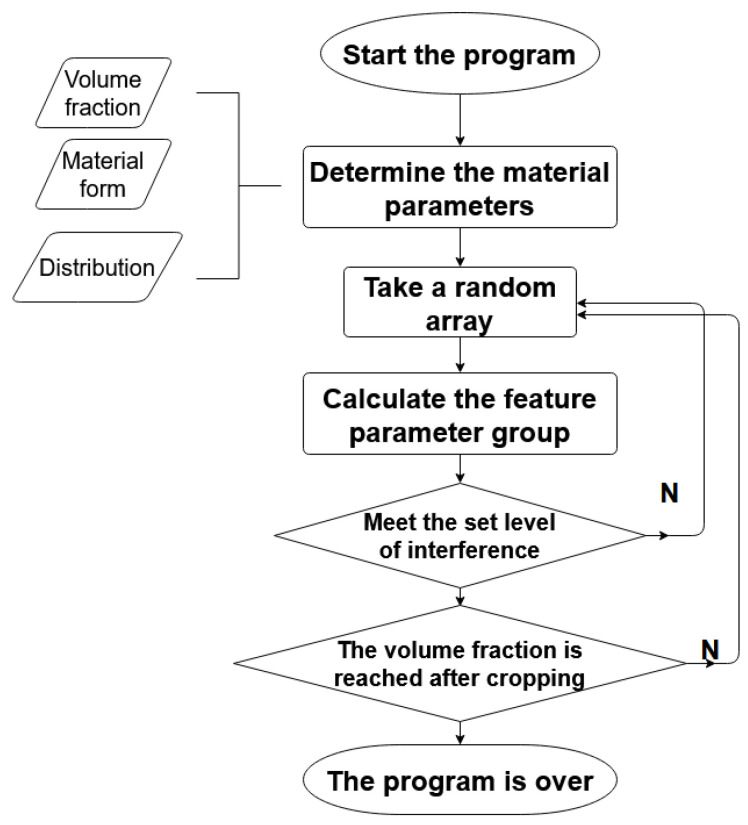
Automatically generated algorithm flowcharts for RVE models.

**Figure 3 materials-14-07656-f003:**
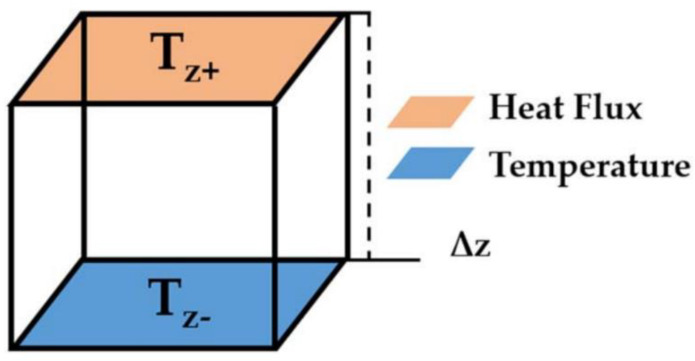
Constant heat flux and constant temperature were set on the two opposite faces of RVE [30].

**Figure 4 materials-14-07656-f004:**
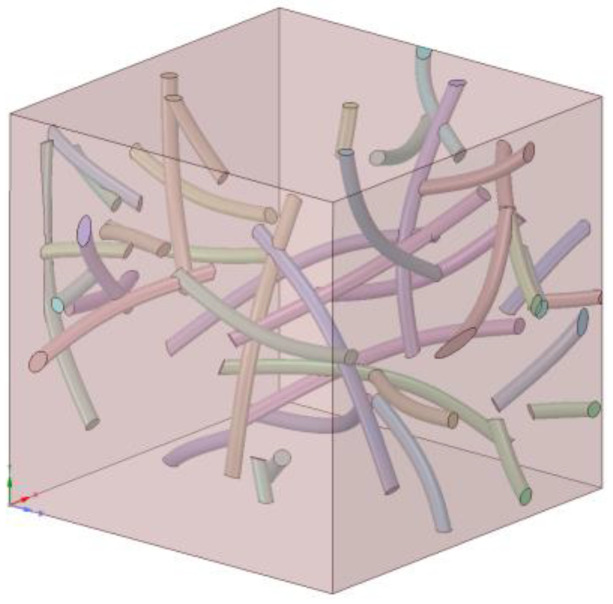
A model of randomly distributed CNTs at a loading of 2% vol.

**Figure 5 materials-14-07656-f005:**
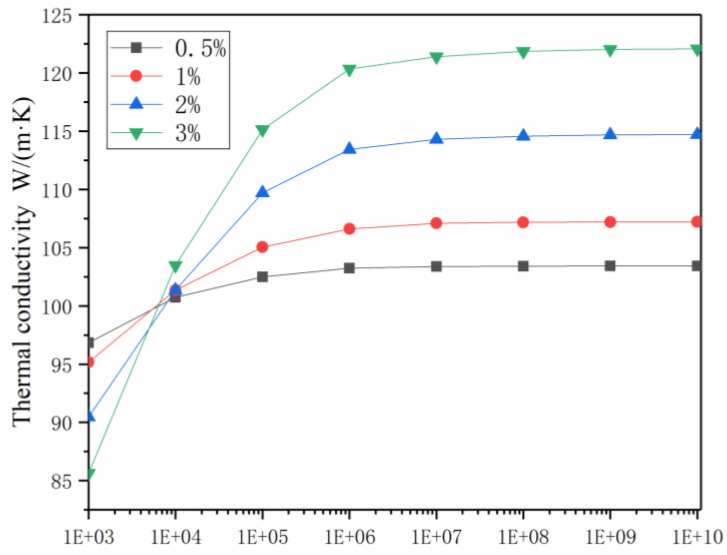
The k_c_ values of CNTs/Co models with randomly distributed CNTs as a function of the C_C-Co_ at CNT loadings of 0.5% vol., 1% vol., 2% vol., 3% vol.

**Figure 6 materials-14-07656-f006:**
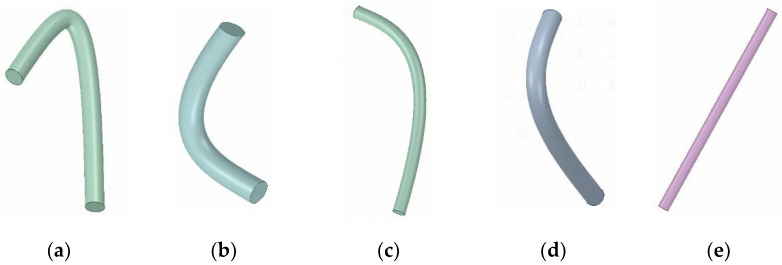
Bending angles of different carbon nanotube models: (**a**) 30 degree angle; (**b**) 60 degree angle; (**c**) 100 degree angle; (**d**) 140 degree angle; (**e**) 180 degree angle.

**Figure 7 materials-14-07656-f007:**
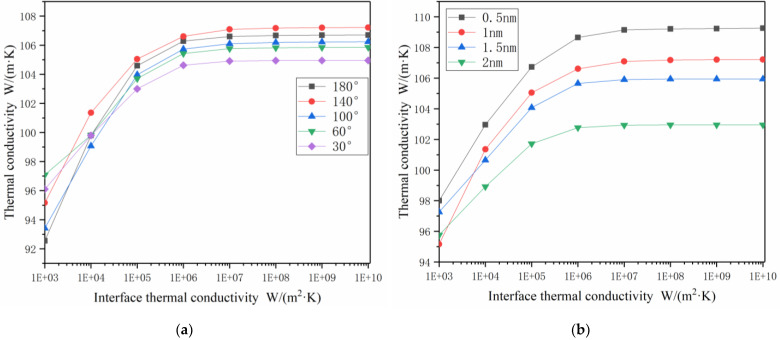
Impact of different influencing factors on k_c_: (**a**) The k_c_ values of CNTs/Co models with different degrees of bending CNTs; (**b**) The k_c_ values of CNTs/Co models with different radii of CNTs.

**Figure 8 materials-14-07656-f008:**
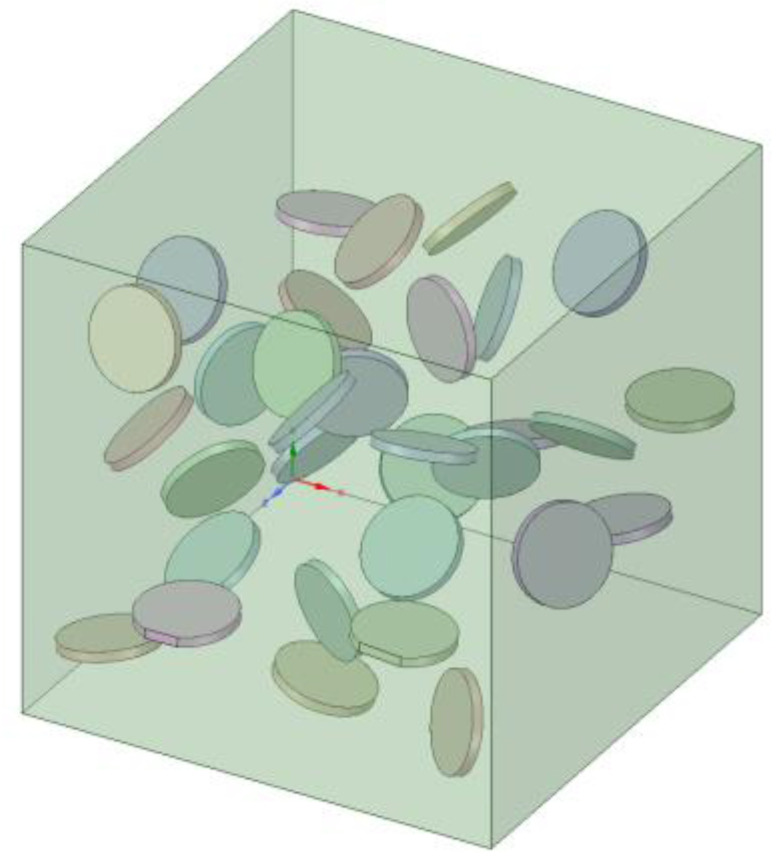
A model of randomly distributed GNPs at a loading of 2% vol.

**Figure 9 materials-14-07656-f009:**
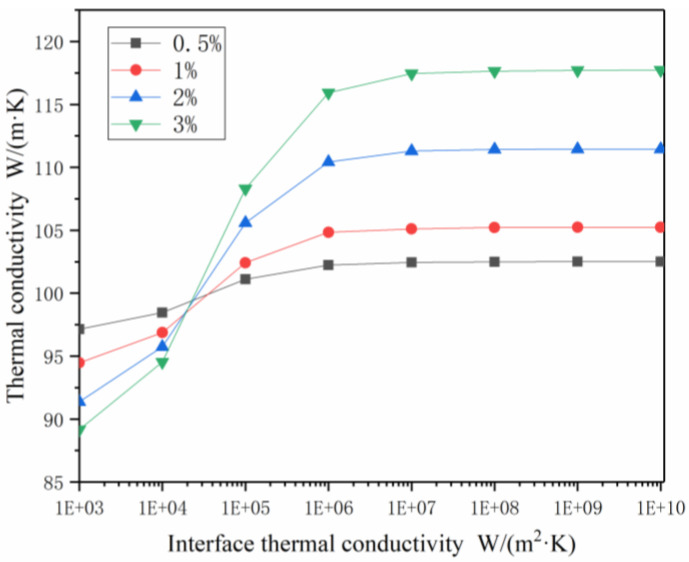
The k_c_ values of GNPs/Co models with randomly distributed GNPs as a function of the C_G-Co_ at GNP loadings of 0.5% vol., 1% vol., 2% vol., 3% vol.

**Figure 10 materials-14-07656-f010:**
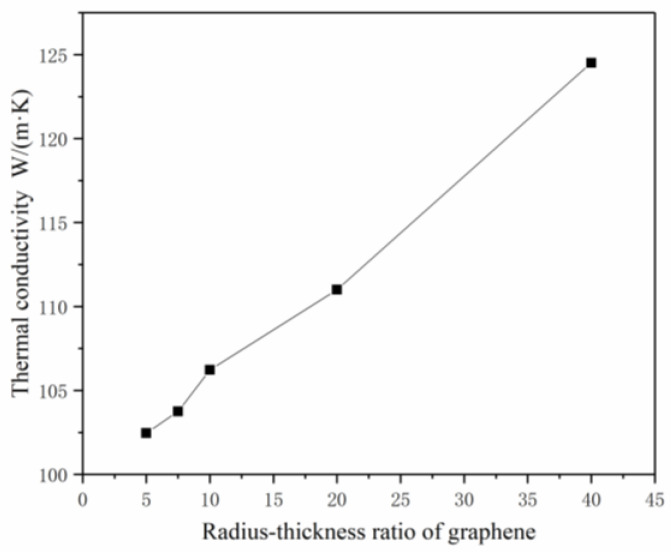
The effects of the diameter–thickness ratio of graphene on K_c_.

**Figure 11 materials-14-07656-f011:**
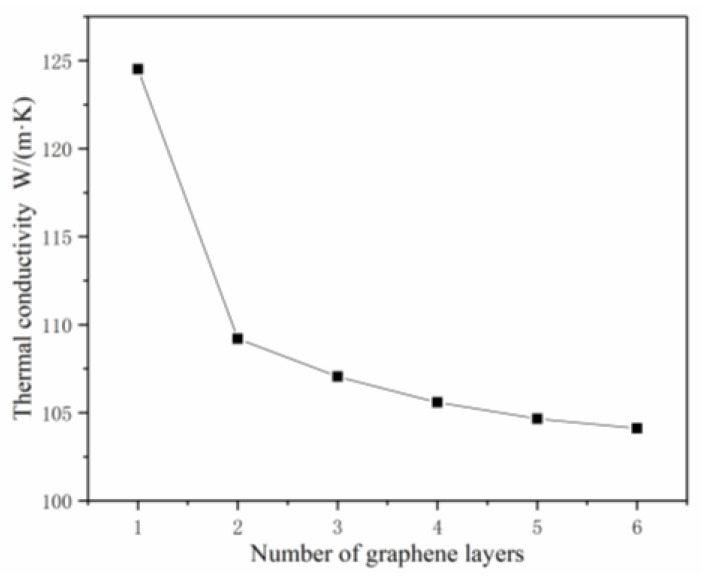
The K_c_ of models of graphene with different degrees of agglomeration.

**Figure 12 materials-14-07656-f012:**
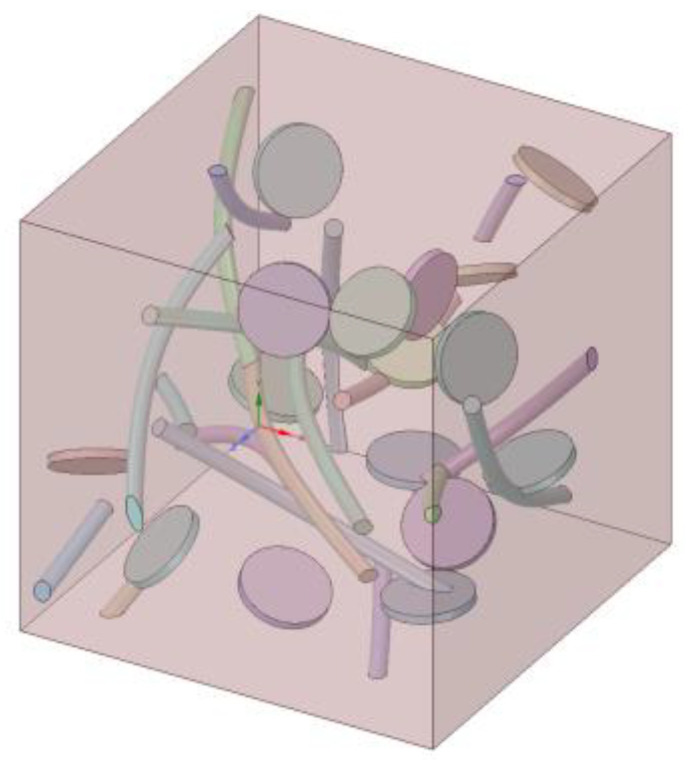
A model of randomly distributed 1% vol. GNPs and 1% vol. CNTs.

**Figure 13 materials-14-07656-f013:**
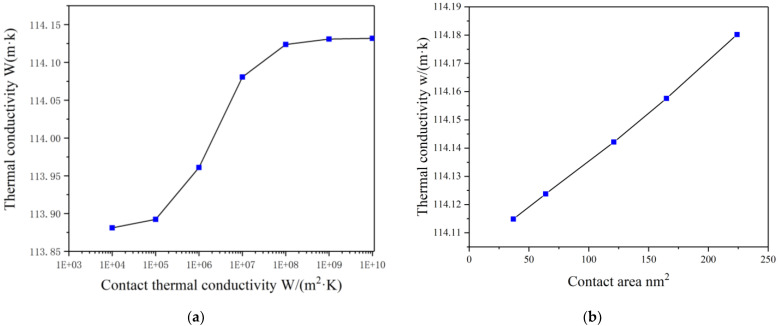
Impact of different influencing factors on K_c_ (**a**) The K_c_ values of models of different C_G-C_ values; (**b**) The K_c_ values of different contact areas between graphene and carbon nanotubes.

**Figure 14 materials-14-07656-f014:**
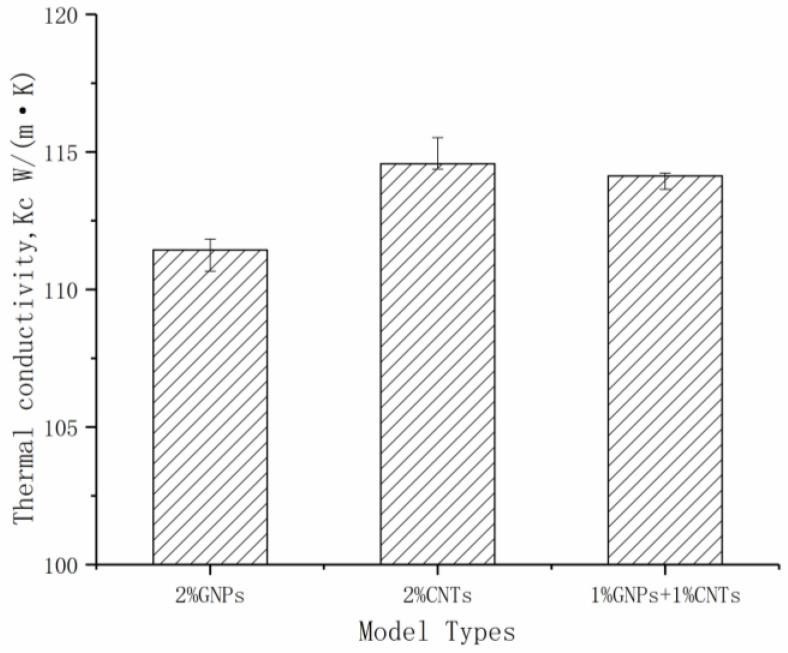
The K_c_ values of three different addition phase models.

**Table 1 materials-14-07656-t001:** FEM simulation of the thermal conductivity of the materials used.

Material	Thermal Conductivity (W/(m·K))
Graphene	5000
Carbon nanotubes	4000
Co	100

**Table 2 materials-14-07656-t002:** The composition of the test samples.

Sample Group Number	GNP (vol.%)	CNT (vol.%)
N	0	0
G1	0.06	0
G2	0.12	0
G3	0.18	0
C1	0	0.06
C2	0	0.12
C3	0	0.18
G1-C1	0.06	0.06

**Table 3 materials-14-07656-t003:** Calculation results of thermal conductivity of four models.

Dispersion State	D_0.1_	D_0.2_	Thermal Conductivity W/(m·K)
**Random distribution**	0.245	0.557	115.7
**Agglomeration** **(Four in a group)**	0.198	0.445	113.2
**Agglomeration** **(Ten in a group)**	0.158	0.288	109.5
**Agglomeration** **(All CNTs)**	0.033	0.109	97.6

**Table 4 materials-14-07656-t004:** The composition of the test samples.

Model Number	L_G_ (nm)	L_C_ (nm)	L (nm)	Thermal Conductivity (W/(m·K))
1	14.775	9.451	24.226	114.88
2	12.588	13.774	26.362	115.26
3	17.665	11.56	29.225	115.89

**Table 5 materials-14-07656-t005:** Thermal diffusivity of WC–Co cemented carbide.

Sample Number	N	G1	G2	G3	C1	C2	C3	G1-C1
**Thermal Diffusivity** **(mm^2^/s)**	28.0	28.33	29.71	29.68	28.18	28.60	28.86	29.30

**Table 6 materials-14-07656-t006:** Density of WC–Co cemented carbide.

Sample Number	Mass (g)	Volume (cm^3^)	Density (g/cm^3^)
N	2.81	0.19	14.80
G1	2.31	0.16	14.44
G2	2.33	0.16	14.55
G3	2.86	0.20	14.32
C1	2.79	0.19	14.70
C2	2.49	0.17	14.67
C3	2.74	0.19	14.44
G1-C1	3.10	0.21	14.75

**Table 7 materials-14-07656-t007:** Specific heat capacity of WC–Co cemented carbide.

Sample Group Number	Specific Heat Capacity J/(g·°C)
N	0.3730
G1	0.4209
G2	0.4086
G3	0.4207
C1	0.3733
C2	0.4034
C3	0.4161
G1-C1	0.3995

**Table 8 materials-14-07656-t008:** Thermal conductivity of WC–Co cemented carbide.

Sample Group Number	Thermal Conductivity (W/(m·K))
N	154.6
G1	172.1
G2	176.6
G3	178.8
C1	154.6
C2	169.3
C3	173.4
G1-C1	172.6

## Data Availability

The data presented in this study are available on request from the corresponding author.

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
