# Peer review of "A Finite Element Analysis of the Effects of Graphene and Carbon Nanotubes on Thermal Conductivity of Co Phase in WC–Co Carbide"

_materials, 2021, doi:10.3390/ma14247656_

Round 1

Reviewer 1 Report

The reviewer comments of the paper «A finite element analysis of the effects of graphene and carbon nanotubes on thermal conductivity of WC-Co carbide»- Reviewer

The authors presented an article «A finite element analysis of the effects of graphene and carbon nanotubes on thermal conductivity of WC-Co carbide». However, there are several points in the article that require further explanation.

Comment 1:

The abstract needs to be improved.

Demonstrate in the abstract novelty, practical significance. Add quantitative and qualitative work results to the abstract. Add an FME method error.

Comment 2:

The introduction needs to be improved.

Firstly, group quotation is unacceptable in one phrase, for example [4-6], [7-10], [18-20], [21-23]. Break this sentence into parts or individual sentences. For example, ... [...], ... [...], etc. Or one reference - one sentence.

Now the list of references needs to be supplemented with at least 3-4 more references published over the past 5 years. Here are some recent article:

Journal of Materials Research and Technology 2021, 15, 327–341. doi:10.1016/j.jmrt.2021.08.037

After analyzing the literature, show before formulating the goal of the "blank" spots. Which has not been previously done by other researchers. You must show the importance of the research being undertaken. Show what will be the new research approach in this article. You need to show a hypothesis.

Add a clear purpose to the article.

Comment 3:

  1. Materials and Methods

Are all figures original? If not needed appropriate citations and permissions. Refine this for figures throughout the article.

The quality and resolution of all figures needs to be improved.

Are all formulas original? If not needed appropriate citations.

Describe the measurement procedure in more detail. At what point in time? How is the measuring setup set up? How many repetitions of measurements? What statistical methods are used to process experimental results? Describe the experimental stand in more detail. What method of experiment planning is used and why?

Finite Element modeling should be described and illustrated in more detail.

Add a design diagram with boundary conditions on the shape. Justify your choice and assumptions made. What type of finite elements are used and why? How does the size of finite elements affect the accuracy of calculations? What parameters did you calculate using the FEM and why?

Give the parameters of the PC on which the forces were calculated using the FEM. How much time was spent on calculations? It is useful to give such parameters in the table and briefly explain the performance in the text.

Comment 4:

  1. Results and discussion

Figures 16,17,18 should be redrawn in color.

Is it important to show how the validation of the FEM models was assessed?

Comment 5:

It will be useful to add a section of Nomenclature in which to sign all the physical quantities and abbreviations encountered in the article. There are many physical quantities in the text and such a section will help to find the description of the necessary element.

For example,

  • : Density (g/cm3)

FEM         : Finite Element Modelling

etc.

Use "rpm" instead of "r/min".

Check it out everywhere in the text.

Comment 6:

Conclusions should be rewritten.

It is necessary to more clearly show the novelty of the article and the advantages of the proposed method. Add qualitative and quantitative results of your work. What is the error of the obtained models? What is the difference from previous work in this area? Show practical relevance. Conclusions should reflect the purpose of the article.

Use the format:

  • Conclusions 1
  • Conclusions 2
  • Etc.

The article is interesting, but needs to be improved. Authors should carefully study the comments and make improvements to the article step by step. After major changes can an article be considered for publication in the "Materials".

Author Response

Reply Letter

Dear Reviewers,

Thank you very much for your suggestions on my article, I have revised them one by one. I will respond to your questions one by one below.

Comment 1:

The abstract needs to be improved.

Demonstrate in the abstract novelty, practical significance. Add quantitative and qualitative work results to the abstract. Add an FME method error.

Re: I have rewritten the abstract and revised it to address the issues you raised, and you can see the new abstract in the attachment.

“In engineering practice, the service life of cemented carbide shield tunneling machine in uneven soft and hard strata will be seriously reduced due to thermal stress. When Carbon Nanotubes (CNTs) and Graphene Nano Platelets (GNPs) are added to WC-Co carbide as enhanced phases, the thermal conductivity of carbide is significantly improved. Researches should be done to further understand the mechanism of enhancement in composites and find ways to assist the design and optimization of the structure. In this paper, a series of finite element models were established using scripts to find the factors that affect the thermal conduction, including positions, orientations, interface thermal conductivity, shapes, sizes and so on. WC-Co carbide with CNTs (0.06%, 0.12%, and 0.18%vol), GNPs (0.06%, 0.12%, and 0.18%vol) and hybrid CNTs-GNPs (1:1) were prepared to verify the reliability of finite element simulation results. The results show that the larger the interface thermal conductivity, the higher the composite phase thermal conductivity. Each 1%vol of CNTs increases the thermal conductivity of the composite phase by 7.2%, and each 1%vol of GNPs increases the thermal conductivity of the composite phase by 5.2%. Proper curvature (around 140°) of CNTs and GNPs with proper diameter to thickness ratio would have better thermal conductivity.”

Comment 2:

The introduction needs to be improved.

Firstly, group quotation is unacceptable in one phrase, for example [4-6], [7-10], [18-20], [21-23]. Break this sentence into parts or individual sentences. For example, ... [...], ... [...], etc. Or one reference - one sentence.

Now the list of references needs to be supplemented with at least 3-4 more references published over the past 5 years. Here are some recent article:

Journal of Materials Research and Technology 2021, 15, 327–341. doi:10.1016/j.jmrt.2021.08.037

After analyzing the literature, show before formulating the goal of the "blank" spots. Which has not been previously done by other researchers. You must show the importance of the research being undertaken. Show what will be the new research approach in this article. You need to show a hypothesis.

Add a clear purpose to the article.

Re: The introductory section has been revised as you suggested. The innovative nature of the article has been made clearer and a clearer experimental purpose has been given. References (Journal of Materials Research and Technology) from the last five years have been added as you suggested. More details can be found in the attachment.

Comment 3:

  1. Materials and Methods

Are all figures original? If not needed appropriate citations and permissions. Refine this for figures throughout the article.

The quality and resolution of all figures needs to be improved.

Are all formulas original? If not needed appropriate citations.

Describe the measurement procedure in more detail. At what point in time? How is the measuring setup set up? How many repetitions of measurements? What statistical methods are used to process experimental results? Describe the experimental stand in more detail. What method of experiment planning is used and why?

Finite Element modeling should be described and illustrated in more detail.

Add a design diagram with boundary conditions on the shape. Justify your choice and assumptions made. What type of finite elements are used and why? How does the size of finite elements affect the accuracy of calculations? What parameters did you calculate using the FEM and why?

Give the parameters of the PC on which the forces were calculated using the FEM. How much time was spent on calculations? It is useful to give such parameters in the table and briefly explain the performance in the text.

Re: All figures are original. Formulas (1-5) are not original and need to be cited, and formulas (6-8) are original.

The measurement process has been described more similarly in "Materials and Methods ", please see the attachment for details.

The detailed steps including measurement instruments, measurement methods, number of measurements, and FEM model building are all explained in detail in the revised manuscript.

The cell types, cell sizes and environmental conditions used in the finite element simulations have been supplemented in "3.1. FEM simulation ".

The quality and resolution of all the figures have been improved to the best of my ability. The images obtained using the screenshot tool are clearer now.

Comment 4:

  1. Results and discussion

Figures 16,17,18 should be redrawn in color.

Is it important to show how the validation of the FEM models was assessed?

Re: The Figures have been redrawn. We believe it is important to show the validation of the FEM models. Because, any model of FEM is not a model set up in isolation from the real situation. The results obtained from the simulation can be convincing only if the model is based on the realistic situation and after verifying its reliability. We built a model of WC-Co for secondary simulations after conducting preliminary simulation experiments. Corresponding this data to the reference, the reliability of the model was confirmed. Due to the space problem, we consider it sufficient that the trend of the sintering experimental results is similar to the trend of the simulation experimental results.

  1. Chen, K.;Xiao, W. K.;Li, Z. W.;Wu, J. S.;Hong, K. R.;Ruan, X. F. Effect of Graphene and Carbon Nanotubes on the Thermal Conductivity of WC-Co Cemented Carbide. Metals. 2019;9(3)

Comment 5:

It will be useful to add a section of Nomenclature in which to sign all the physical quantities and abbreviations encountered in the article. There are many physical quantities in the text and such a section will help to find the description of the necessary element.

For example,

  • : Density (g/cm3)

FEM         : Finite Element Modelling

etc.

Use "rpm" instead of "r/min".

Check it out everywhere in the text.

Re: Already changed

Comment 6:

Conclusions should be rewritten.

It is necessary to more clearly show the novelty of the article and the advantages of the proposed method. Add qualitative and quantitative results of your work. What is the error of the obtained models? What is the difference from previous work in this area? Show practical relevance. Conclusions should reflect the purpose of the article.

Use the format:

  • Conclusions 1
  • Conclusions 2
  •  

Re: Already changed

“In this paper, a programmable controlled FEM model with random distribution of the added phases in space was innovatively developed, and the factors affecting the mixing of carbon nanotubes and graphene on the thermal conductivity of WC-Co cemented carbide were discussed. Sintering experiments were designed to verify the correctness of the results of FEM in terms of trend. It can be seen from the results of FEM simulation that:

  1. The better the dispersion of CNTs, the larger the volume fraction and the smaller the diameter and the greater the interfacial thermal conductivity with CO phase, the higher the thermal conductivity of composites. Each 1%vol of CNTs increases the thermal conductivity of the composite phase by 7.2%. An appropriate degree (about 140°) of bending will provide better result.
  2. The better the dispersion of GNPs, the larger the volume fraction and the ratio of diameter to thickness and the greater the interfacial thermal conductivity with CO phase, the higher the thermal conductivity of composites. Each 1%vol of GNPs increases the thermal conductivity of the composite phase by 5.2%. The dispersed graphene creates a wider range of heat flow networks in space.
  3. When CNTs and GNPs were incorporated into the Co phase, the magnitude of the contact thermal conductivity CC-G has less effect on the thermal conductivity of the composite phase due to the small contact area between CNTs and GNPs.

All of the above factors can be improved to increase the thermal conductivity of carbide with the addition of CNTs and GNPs, which was verified in sintering experiments.”

Reviewer 2 Report

The paper deals with the FE modeling of thermal conductivity in Co-CNTS and Co-graphene composites. The topic of the paper is interesting. However the presentation of the results is confusing. It seems that the paper is written be different author and therefor some parts are well written and others are very poor written. Therefore, the structure of the paper is not well prepared making the manuscript difficult to be read.

Several remarks should be addressed and clarified in order for the paper to be publishable.

Experimental section/results/discussion

  1. The author model a Co matrix with graphene and CNTs enhacing phases. However the title refers to WC-Co matrix. The authors should properly adjust the title.
  2. The authors should explain if an RVE of 50 nm3 is adequate for obtaining realistic results. It seems to me that the size of the RVE is very small.
  3. The authors should explain the exact conditions of modelling. What type of elements they used for each phase. What type of contact between each phase? What is the bonding between the interface and the other phases, etch
  4. Thermal (Temperature distributions) and heat flux distribution patterns are not depicted. The authors are advised to include such results.
  5. Figure 5 and section 2.2.5: The other surfaces are kept adiabatic to ensure that the heat will flow in one direction?
  6. Literature survey should be more emphasized in modeling techniques for thermal properties of graphene and carbon nanotubes composites. The authors are advised to add such literature especially in the results and discussion section so as to give better explanation of their results and correlate their results with other published works.
  7. It is not clear why the interface thermal conductivity is different for the carbon nanotubes models. The same applies for the graphene models. The authors try to explain that in page 3 but they refer to molecular dynamics modeling. In continuum mechanic such explanation referring to phonons cannot be employed. Is there appearance of other phases in the interface? Is it related with the bonding of the reinforcement phase and the matrix (eg. a perfect strong bonding or a weak or less perfect bonding?). Moreover, the authors in lines 281-290 relate the interface conductivity of CNTs with their purity and defects. However, do the authors use same quality and purity of nanotubes in each model which is the same as the experimental material? Consequently, this reviewer does not understand the data presented in Fig 7, 9, 10, 12 and 16.
  8. The validation of the modeling is not clear. A direct comparison should be added preferably in a graphical manner. Moreover the author use WC-Co cemented carbides but in their models they use only Co as the matrix phase. The authors should clarify this discrepancy.

9              The authors should try to reduce the figures. They should try to group the representation of their results. For example figure 1, 2 and 3 can be combined in one figure with several sections.

General

  1. Extensive editing should be done in the manuscript. Same examples (but are limited to these) are in line 7à sons, line 123à CTNS, line 277à error! , etch.
  2. The last paragraph of the introduction section should be rewritten. The English usage is very poor and is difficult to comprehend.
  3. The section 2.3.1 and 2.3.2 should also be rewritten. The English usage is very poor and is difficult to comprehend.
  4. The titles of section 2.1, 2.2.4 should be changed appropriately, For example 2.1 Simplified finite element model.

Author Response

Reply Letter

Dear Reviewers,

Thank you very much for your suggestions on my article, I have revised them one by one. I will respond to your questions one by one below.

Experimental section/results/discussion

  1. The author model a Co matrix with graphene and CNTs enhacing phases. However the title refers to WC-Co matrix. The authors should properly adjust the title.
  2. The authors should explain if an RVE of 50 nmis adequate for obtaining realistic results. It seems to me that the size of the RVE is very small.
  3. The authors should explain the exact conditions of modelling. What type of elements they used for each phase. What type of contact between each phase? What is the bonding between the interface and the other phases, etch
  4. Thermal (Temperature distributions) and heat flux distribution patterns are not depicted. The authors are advised to include such results.
  5. Figure 5 and section 2.2.5: The other surfaces are kept adiabatic to ensure that the heat will flow in one direction?
  6. Literature survey should be more emphasized in modeling techniques for thermal properties of graphene and carbon nanotubes composites. The authors are advised to add such literature especially in the results and discussion section so as to give better explanation of their results and correlate their results with other published works.
  7. It is not clear why the interface thermal conductivity is different for the carbon nanotubes models. The same applies for the graphene models. The authors try to explain that in page 3 but they refer to molecular dynamics modeling. In continuum mechanic such explanation referring to phonons cannot be employed. Is there appearance of other phases in the interface? Is it related with the bonding of the reinforcement phase and the matrix (eg. a perfect strong bonding or a weak or less perfect bonding?). Moreover, the authors in lines 281-290 relate the interface conductivity of CNTs with their purity and defects. However, do the authors use same quality and purity of nanotubes in each model which is the same as the experimental material? Consequently, this reviewer does not understand the data presented in Fig 7, 9, 10, 12 and 16.
  8. The validation of the modeling is not clear. A direct comparison should be added preferably in a graphical manner. Moreover the author use WC-Co cemented carbides but in their models they use only Co as the matrix phase. The authors should clarify this discrepancy.

9              The authors should try to reduce the figures. They should try to group the representation of their results. For example figure 1, 2 and 3 can be combined in one figure with several sections.

Re:

  1. First of all, I am very glad to see this question, which shows that you have seriously considered what the main body of the article should be. However, I personally do not think the title of the article needs to be changed for the following reasons. “WC-Co carbide is made by bonding the hard phase WC and the bonding phase Co. The addition of GNPs and CNTs enhances the thermal conductivity of cemented carbide. However, GNPs and CNTs are not uniformly distributed throughout the cemented carbide, but are concentrated in the Co phase or on the contact surface between the Co phase and WC. So the FEM model for making WC-Co carbide can be simplified to make a finite element model of the Co phase with the addition of GNPs and CNTs.” This can be seen in "2.1 The FEM model is simplified ". Secondly, the samples for sintering experiments were carbide samples with GNPs and CNTs added. Finally, the research topic of this study is the carbide failure of shield inserts. A link to real life is the WC-Co carbide. Therefore, we believe that the title of the article does not need to be changed.
  2. An RVE of 50 nmis adequate for obtaining realistic results. Professor Xiao's students have done simulations on this[1, 2] .This shows that 50 nm3 of RVE model is able to give the corresponding results.
  3. The measurement process has been described more similarly in "Materials and Methods ", please see the attachment for details.The detailed steps including measurement instruments, measurement methods, number of measurements, and FEM model building are all explained in detail in the revised manuscript.The cell types, cell sizes and environmental conditions used in the finite element simulations have been supplemented in "3.1. FEM simulation ".The quality and resolution of all the figures have been improved to the best of my ability. The images obtained using the screenshot tool are clearer now.
  4. This study was conducted to quantitatively explore carbon nanotube volume fractions, carbon nanotubes and Co phases Effect of interface thermal conductivity and carbon nanotube form on composite Co-phase thermal conductivity, effect of graphene volume fraction, graphene and Co-phase interface thermal conductivity, graphene distribution on composite Co-phase thermal conductivity, and effect of graphene and carbon nanotubes on composite phase thermal conductivity.We have experimental results on heat (temperature distribution) and heat flux distribution patterns, but given the purpose of the article and the length of the article, I don't think it is necessary to add them.
  5. Sorry, I missed this information. Yes, other surfaces are kept adiabatic to ensure that heat flows in one direction.
  6. We are happy to accept your comments and have made the appropriate changes to address this issue, which can be seen in detail in the attached document.
  7. Thank you very much for your comments, it was an oversight in my writing process. “In continuum mechanic such explanation referring to phonons cannot be employed.” This is correct. We use FEM to avoid this difficulty. Details of the changes to this section can be found in the "Materials and Methods" section. The quality and purity of the nanotubes used in the model must not be the same as the experimental material. This is because the experimental materials used in real life are unlikely to follow a certain data from the simulation. The purpose of the sintering experiments is simply to determine the reliability of the trends in the simulation results. It is because the type of carbon nanotube affects its thermal conductivity that its thermal conductivity is explored in terms of volume fraction, degree of bending and radius. The specific changes are also reflected in the article. The reasons for the influencing factors are described in more detail.
  8. Like I said in reply 1.
  9. It has been modified as far as possible.

A revised manuscript has been submitted in the attached document for your further review.

[1].  Wang, H.;Xiao, E.;Fan, T.;Li, X.;Xiao, W. Calculations of factors that affect thermal conductivity in epoxy composites with hybrid carbon nanotube and graphene nano platelet. Materials Research Express. 2020;7(2).doi:10.1088/2053-1591/ab71ca

[2].   Xiao, W.;Zhai, X.;Ma, P.;Fan, T.;Li, X. Numerical study on the thermal behavior of graphene nanoplatelets/epoxy composites. Results in Physics. 2018;9:673-679.doi:10.1016/j.rinp.2018.01.060

Round 2

Reviewer 1 Report

The authors have improved the article. However, add 3-4 more references as before. Including the suggested article earlier:
Journal of Materials Research and Technology 2021, 15, 327–341. doi:10.1016/j.jmrt.2021.08.037
After that, the article can be accepted for publication.

Author Response

Thank you very much for this reference, which I have added to the article.

Personally, I think the number of references is sufficient.

I very much hope that the article will be accepted well after this change.

Reviewer 2 Report

No futher suggestion. Mayde a small chande in the title "WC-Co cemented carbide"

Author Response

Thank you very much for your persistence in this proposal.

After carefully rethinking your suggestion, I decided to make a slight change to the title.

A finite element analysis of the effects of graphene and carbon nanotubes on thermal conductivity of Co phase in WC-Co carbide

In this way, it can take into account the ideas of both of us.

Round 3

Reviewer 1 Report

There are no more comments.